# Exotic Species and Autochthonous Parasites: Trichostrongylus Retortaeformis in Eastern Cottontail

**DOI:** 10.3390/life10040031

**Published:** 2020-03-25

**Authors:** Chiara Gontero, Angela Fanelli, Stefania Zanet, Pier Giuseppe Meneguz, Paolo Tizzani

**Affiliations:** Department of Veterinary Sciences, University of Turin, 10090 Grugliasco (TO), Italy; chiagonti@gmail.com (C.G.); angela.fanelli@unito.it (A.F.); stefania.zanet@unito.it (S.Z.); piergiuseppe.meneguz@unito.it (P.G.M.)

**Keywords:** lagomorphs, *Sylvilagus floridanus*, invasive species, *Trichostrongylus retortaeformis*

## Abstract

Introduction: A parasite community is usually well adapted and specific to the host species they co-evolved with. Although exotic pathogens infecting autochthonous species have been documented, the infection of an alien species with native parasites is rare in lagomorphs. *Trichostrongylus retortaeformis* is a nematode parasite infecting the small intestine of domestic and wild lagomorphs in Europe. Methods: Thirty-two Eastern cottontails from a naturalized population in Italy were processed to describe the gastrointestinal parasite community. Results and discussions: *T. retortaeformis* is reported for the first time in the Eastern cottontail *Sylvilagus floridanus* introduced to Europe. The Eastern cottontail is an invasive lagomorph, living in sympatry with the autochthonous European brown hare in certain areas of Italy. This study provides new insights into the dynamics of parasite communities of native and alien lagomorph species in sympatric areas.

## 1. Introduction

An ecosystem is defined as a network of biotic (e.g., plants, animals, and microorganisms) and abiotic factors (e.g., soil, water, air, and climate), interacting as a single functional unit [1]. Animals and plants occupy a specific ecological niche, in equilibrium with the other elements. When allochthonous species are released in a new ecosystem, the state of equilibrium is no longer maintained [2]. An allochthonous species, able to reproduce and to colonize a new environment, is defined as an “invasive alien species”, which can threaten biodiversity [2]. Although the risks associated with the introduction of aliens have been extensively documented, each year several allochthonous species are released outside their native range [2]. From 2005 to 2008, the European project DAISIE (Delivering Alien Invasive Species Inventories for Europe) recorded more than 11,000 invasive species in Europe (www.europe-aliens.org). It is estimated that 15% of these have caused economic damage, and another 15% have threatened biodiversity [2]. Eight hundred thirty-three invertebrate species, 1007 plants, and 76 terrestrial vertebrates have been introduced to Italy (www.europe-aliens.org). Alien introductions may also involve parasites, bacteria, or viruses [3,4]. Indeed, the risk of introducing pathogens through animals not adequately controlled from a sanitary point of view is likely [5]. Since the 1950s, following the reduction of the European brown hare *Lepus europaeus* [6], the Eastern cottontail *Sylvilagus floridanus* was introduced for hunting purposes in several European countries: France (1953), Italy (1966), Spain (1980), and Switzerland (1982) [7].

The only population that successfully established itself in Europe was the Italian one [8], being of particular biological interest since it is in sympatry with the native European brown hare [9]. Indeed, it has been demonstrated that the invasive Eastern cottontail poses a risk to native lagomorphs by introducing exotic parasites like *Obeliscoides cuniculi* or *Trichostrongylus affinis* [10,11,12]. Despite the fact that exotic pathogens infecting autochthonous species are frequently described in the literature [3,5,11,13], the infection of an alien species with native parasites is less frequent [14,15], at least in lagomorphs [13]. The spill-over of autochthonous parasites to invasive species is occasionally reported, either due to infection with low numbers of parasites or to limited diagnostic capabilities [15]. The aim of this study is to assess if Eastern cottontail living in sympatry with the European brown hare population can acquire parasites from autochthonous lagomorphs [16]. Moreover, different hypotheses (reduce fitness, dominance, host density) are explored to evaluate the structure of the parasite community.

## 2. Material and Methods

The study was carried out in the protected area “Tortona-Rivalta” (44.86 latitude, 8.82 longitude; Piedmont, Italy) where *L. europaeus* and *S. floridanus* share the same areas [8]. Thirty-two *S. floridanus*, hunted during population control programs, were necropsied. The intestines were examined for the presence of parasites. Standard procedures for isolation and identification of nematodes were implemented [17]. A minimum number of 15 males, if available, was examined for each infected animal under a light microscope with a magnification factor up to 400×. Identification was carried out using the key provided by Skrjabin [18] in order to evaluate the composition of the parasite community. Parasite prevalence (P), abundance, intensity, and ratio between parasite species (in case of mixed infection) were recorded. The infection ratio was recorded as a percentage of *T. retortaeformis* males in comparison with the total number of parasite males identified in each sample.

All applicable international, national, and/or institutional guidelines for the care and use of animals were followed.

## 3. Results and Discussion

Two species of nematodes were found: *Trichostrongylus retortaeformis* (T.R.) and *Trichostrongylus calcaratus* (T.C.) (Figure 1). Specimens of *T. retortaeformis* and *T. calcaratus* have been deposited at the Museum of Natural History of Carmagnola (Turin–Italy), with catalog number MCCI/100-119.

Prevalences, 95% confidence intervals, combined abundance (of the two parasites), and species ratio (for mixed infection) are reported in Table 1. *T. retortaeformis* was identified in 2 Eastern cottontails (6%, CI_95%_ = 0–14%), while *T. calcaratus* was recorded in 96.6% (CI_95%_= 90–100%) of the samples.

*T. retortaeformis* is one of the most common nematodes of several European lagomorphs, including *L. europaeus* [19], *Lepus timidus varronis* [20,21], *Lepus timidus scoticus* [18], and *Oryctolagus cuniculi* [22,23,24]. The parasite is quite a generalist and is able to infect species other than lagomorphs, including ruminants, rodents, and possums [25,26,27] On the other hand, *T. calcaratus* is the common intestinal nematode of Eastern cottontail, reported in Italy by Meneguz and Tizzani [10]. To our knowledge, this is the first report of *T. retortaeformis* in *S. floridanus*, and the first time that *S. floridanus* is infected by nematodes different from the ones characterizing its parasite community. Prior to our studies, allochthonous parasites introduced with the Eastern cottontail have been found to infect the European brown hare in Italy [13,28]. However, the infection of invasive cottontails with autochthonous parasites has never been documented before. *T. retortaeformis* was found at low prevalence and abundance, in comparison with the values registered in its natural hosts [16,22]. In our sample, *T. calcaratus* remained the dominant parasite species in terms of prevalence and abundance.

There might be several reasons for this unbalanced presence of the two nematodes:Reduced fitness: *T. retortaeformis* may not have fully adapted to its new host, and this may negatively affect its fitness. This hypothesis should be supported by a deeper evaluation of the major indexes of parasite fitness, such as parasite size and spicule length [29].Dominance: *T. calcaratus* and *T. retortaeformis* may compete for the same ecological niche (small intestine). A better adaptation of *T. calcaratus* to its natural host may reduce the prevalence of other parasites (*T. retortaeformis*) [29,30,31].Host population density: *S. floridanus* is present in the study area at densities higher than the European brown hare (ratio 5 to 1). Over the years, the environmental presence of *T. calcaratus* infecting larvae (L3) has probably exceeded the presence of the *T. retortaeformis* L3. This may have led to a progressive reduction of T. *retortaeformis* L3 and adult nematodes in the definitive hosts. The conclusion of this hypothesis is that, in areas of sympatry, *T. retortaeformis* has a higher probability of extinction.

## 4. Conclusions

In conclusion, even if performed on a limited number of individuals, our work highlights how the introduction of allochthonous species can modify ecosystem dynamics with unexpected consequences at both macro and micro scales. The information reported represents a useful case study of the mid and long-term consequences of the uncontrolled translocation of species.

## Figures and Tables

**Figure 1 life-10-00031-f001:**
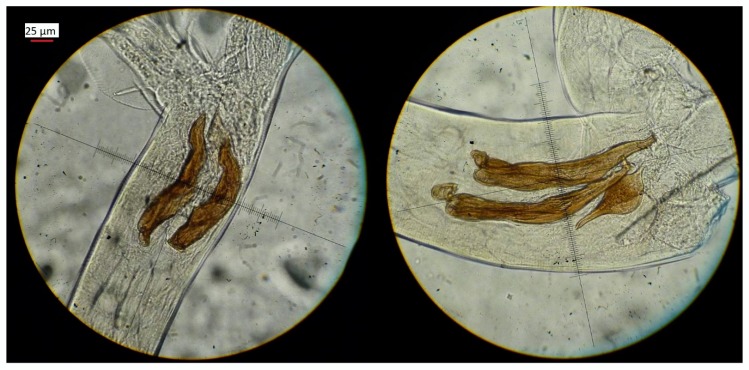
*Trichostrongylus retortaeformis* (on the left) and Trichostrongylus calcaratus (on the right): details of the spicules of the two species (image at 400× magnification).

**Table 1 life-10-00031-t001:** Prevalence, combined abundance, and ratio (for mixed infection) between *T. retortaeformis* (T.R.) and *T. calcaratus* (T.C.).

	Prevalence (%)		Combined Abundance		Spp. Ratio (for Mixed Infection) ♂T.R./♂T.C.
	T.R. (CI_95%_)	T.C. (CI_95%_)	Mean	St. Dev.	
Spring (N = 6)	16.7 (13–46)	100	1058.3	1144.9	16.7%
Summer (N = 9)	0	100	761.1	1039	NA
Autumn (N = 6)	16.7 (13–46)	83.3 (53–100)	138.3	157.9	11.1%
Winter (N = 11)	0	100	31.3	30.5	NA
**Total (N = 32)**	**6.3 (−2–14)**	**96.6 (90–100)**	**449.19**	**821.6**	**13.6%**

Note: “♂ = male”.

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
