# Peer review of "Exotic Species and Autochthonous Parasites: Trichostrongylus Retortaeformis in Eastern Cottontail"

_life, 2020, doi:10.3390/life10040031_

Round 1

Reviewer 1 Report

This manuscript reports new information on an important alien invasive mammal species in Italy, with specific reference to its helminth parasite community.  This is the first report of one important nematode parasite in the introduced host, which is a significant point worthy of publication.  The general questions posed are important not only to understanding this particular host-parasite relationship, but more broadly it is informative regarding the nature of invasive hosts and the parasites they harbor.

Overall, this is a good paper that should be published following substantial revision.  The introduction is well presented; the methods are generally sound, with some problems (see below); the results support the conclusions offered; the relevant literature is satisfactorily reviewed and cited.  The most critical problem in the methods and study design is that the authors do not mention deposition of voucher specimens of the parasites in a major museum collection.  The micrographs of spicules presented are only modestly informative, but do not provide critical confirming information regarding the species, and provide no information on morphological variations that later authors may want to address.  It is imperative that the authors deposit a series of actual nematode specimens from their study in at least one major museum.  This will allow other researchers to corroborate, falsify, or add complementary information at a later time.  Such deposition of voucher specimens should be required for publication of an ecological study.  After depositing the specimens, the authors should include in their paper the name of the museum, and accession catalog numbers of the specimens so that other researchers can retrieve those specimens for verification and further analysis.

The English language of this manuscript is very problematic.  A copy editor can correct some grammar and stylistic errors (e.g., lack of italics in some binomials), and even some poor sentence structure.  However, some of the sentences in this manuscript did not make sense to me, apparently because of poor English language usage.  I strongly urge the authors to seek assistance in correcting and improving the language, in particular to improve accuracy.

If the authors will make these revisions, I think this is a very nice and important study that will be an important contribution to the literature.  Thus, pending deposition of voucher specimens and suitable language revision, I recommend acceptance.

Reviewer 2 Report

This manuscript warrants publication in Life, especially since this is the first report of T. retortaeformis in Sylvilagus floridanus. I think that some revision is required for publication, particularly regarding presentation of the data. Sample sizes are quite low for reasonably accurate rates of the abundance of both species. My comments and suggestions follow.

Line 67. It is not stated what the minimum number (15) of males identified relates to. It could be (1) a host individual (infracommunity), (2) the seasonal sample (component community) or (3) from the results presented in Table 1 (see comment below), some other method. This needs to be clarified.

Table 1. I find the species ratio data confusing. As mentioned in my comment above regarding line 67, the minimum number of males does not seem to be 15 except in summer and winter. If my interpretation of the Spp Ratio column is correct, the total number of males identified is 12 for spring and 9 for autumn. In my opinion, in order to get a reasonably accurate estimate of prevalence and abundance of T. retortaeformis, the aim should have been to reach a minimum target of identified males (say 10) in each host specimen (infracommunity). Obviously this would not have been possible in hosts with very few Trichostrongylus spp. specimens. The table indicates that abundances are generally quite high, so it should not be difficult to identify more males. With more males identified per host there may have been significantly higher prevalence and abundance of T. retortaeformis than presented in this study. Also, the species ratio may be better presented as a percentage (eg 17%:83%).

Table 1. Presumably the Abundance and Intensity both relate to the total number of individuals of both species of Trichostrongylus, whereas Prevalence does not. So it might be better for the column headers to be called "Combined abundance" and "Combined intensity". Also, since the mean abundances differ from mean intensities in just one season and only by a small amount, it might be simpler to just use abundances. The table legend should also be "Prevalence, combined abundance, combined intensity and ratio . . etc.

Figure 1 needs an accurate scale. The statement in the legend (400x magnification) is not correct. The amount of magnification depends on the display size, so if the paper is read on a paper copy it is likely to differ in size to being read online. For example, I printed the manuscript on A4 paper and by dividing the length of a T. retortaeformis spicule by 400 it is only about 75µm long, and T. calcaratus about 110µm. They should be 100-140µm and 170-180µm long respectively. Also, Haakh (1955) noticed that T. retortaeformis was morphologically different in Lepus europaeus compared to Oryctolagus cuniculus. Zeitschrift fur Angewandte Zoologie 2: 151-157. Was there a difference in spicule size in T. retortaeformis in Sylvilagus floridanus compared to those in Lepus europaeus? If so, it could be a useful addition to this paper.

Line 113. Are there any pre-introduction prevalence data available in parts of Italy that now have eastern cottontails? If not, leave this sentence out.

General. As stated in the last sentence of the conclusion, the information reported is of interest to consequences of translocation of species. However, it is really not surprising that T. retortaeformis is able to infect eastern cottontails. It has been reported in a number of hosts other than European lagomorphs, including ruminants, rodents and even possums. Perhaps reference to this should be made in the last paragraph of the introduction.

Trivial comments:

Line 14. Replace "Yet" with "Although"

Line 35. Replace " . . can threat . ." with " . . can threaten . ."

Line 53. Despite the fact that exotic . . .

Line 62. The lat/long figures are far too accurate - 4 decimal places would be plenty, accurate to about 11m.

Line 65. I think it would be better to put the seasonal sample sizes in Table 1 rather than Materials and Methods.

Reviewer 3 Report

Basic reporting,

In this manuscript, the authors provided new information about the dynamics of the parasite communities of native and alien on lagomorph specie.

The paper is very well written, the text is clear and unambiguous. English is correct throughout the paper. The structure of the paper is correct.

However, I would like to comment some points concerning (see attached pdf for the comments). 

Best regards and good luck

Round 2

Reviewer 2 Report

In my opinion the manuscript has been significantly improved and would meet most of the requirements of Reviewer 1, in addition to my own suggestions. So I believe that with a few minor changes, the paper now warrants publication in Life.

My interpretation of the request by Reviewer 1 for the authors to deposit voucher specimens to a museum collection is that both nematode species should be deposited, and I would agree with that. The authors have only deposited specimens of T. retortaeformis.

One point I made in my original review was that the figure needs a scale. This has not been added. Both images in the figure include an ocular micrometer which should be able to be used to calibrate a scale to be added to the figure.

Finally, I think the English language could be improved a little by the following suggestions:

Throughout: Most parasitologists use the terms infect, infection etc. for internal parasites. Infest, infestation etc. usually refers to external parasites.

Line 13: "A parasite community . . "

Line 19: " . . cottontails . .

Line 23: "The Eastern . . "

Line 51: " . . since it is in . . "

Line 63: " . . hypotheses (reduced fitness, dominance, host density) have been . ."

Line 82: " . . T. retortaeformis have . . "

Line 93: " . . nematodes . ."

Line 104: Delete "intensity"

Line 106: " . . in terms of prevalence and abundance." (ie delete "intensity")

Line 109: " . . affect . ."

Line 110: " . . fitness, such as . ."

Line 112: " . . may compete . ."

Reviewer 3 Report

Accepted for publication
